Effects of tropical fruit blends on fermentative and pigmentation aspects of probiotic native cultured goat milk

de Oliveira Galdino Isadora Kaline Camelo Pires isadorakaline@servidor.uepb.edu.br 1 2
da Silva Gabriel Monteiro 3 4
da Silva Miqueas Oliveira Morais 1 5
Dantas Giordanni Cabral 1 5
Pereira Elainy Virgínia dos Santos 5
de Oliveira Tiago Almeida 6
dos Santos Karina Maria Olbrich 7
do Egito Antonio Silvio 8
Alonso Buriti Flávia Carolina 1 5
Cardarelli Haíssa Roberta 9
1 Centro de Ciências Biológicas e da Saúde, Universidade Estadual da Paraíba , Campina Grande , Paraíba , Brazil
2 Centro de Tecnologia–Programa de Pós Graduação em Ciência e Tecnologia de Alimentos, Universidade Federal da Paraíba , João Pessoa , Paraíba , Brazil
3 Departamento de Engenharia Agrícola, Universidade Federal de Campina Grande , Campina Grande , Paraíba , Brazil
4 Departamento de Química, Universidade Estadual da Paraíba , Campina Grande , Paraíba , Brazil
5 Núcleo de Pesquisa e Extensão em Alimentos, Universidade Estadual da Paraíba , Campina Grande , Paraíba , Brazil
6 Departamento de Estatística, Universidade Estadual da Paraíba , Campina Grande , Paraíba , Brazil
7 Embrapa Agroindústria de Alimentos, Empresa Brasileira de Pesquisa Agropecuária , Rio de Janeiro , Brazil
8 Embrapa Caprinos e Ovinos, Núcleo Regional Nordeste, Empresa Brasileira de Pesquisa Agropecuária , Campina Grande , Paraíba , Brazil
9 Centro de Tecnologia e Desenvolvimento Regional, Universidade Federal da Paraíba , João Pessoa , Paraíba , Brazil
Okpala Charles
Electronic publication date: 2025 Jan 13
Publication date: 2025
Volume: 13
Electronic Location ID: e18813
Received 2024 May 9; Accepted 2024 Dec 13
Copyright: ©2025 Kaline Camelo Pires de Oliveira Galdino et al.
Copyright year: 2025
Copyright holder: Kaline Camelo Pires de Oliveira Galdino et al.
License: This is an open access article distributed under the terms of the Creative Commons Attribution License, which permits unrestricted use, distribution, reproduction and adaptation in any medium and for any purpose provided that it is properly attributed. For attribution, the original author(s), title, publication source (PeerJ) and either DOI or URL of the article must be cited.
License URL: https://creativecommons.org/licenses/by/4.0/

Keywords: Lactiplantibacillus plantarum, Acidification, Total phenolic content, Experimental design, Fruit ingredients, Reddish color

Funding: The Post-Graduate Program in Food Science and Technology (PPGCTA/UFPB) FAPESQ Notice no. 4/2021 PDPG from the Semi-Arid Region, Brazilian Agricultural Research Corporation (EMBRAPA, Project PROBIOLACT 23.16.05.038.00.00) Conselho Nacional de Desenvolvimento Científico e Tecnológico (CNPq, Projects 307075/2020-6 and 308253/2020-5) Coordenação de Aperfeiçoamento de Pessoal de Nível Superior (CAPES/PROAP) Fundação de Apoio à Pesquisa do Estado da Paraíba (FAPESQ, Projects 028/2018 and 2778/2023) Fundação Parque Tecnológico da Paraíba (PaqTcPB, Project AC-NUPEA-UEPB) Pró-Reitoria de Pesquisa (PROPESQ/PRPG/UFPB, internal productivity call 03/2020-PVM 13515-2020) Paraíba State University #01/2024 This research had financial support from the Post-Graduate Program in Food Science and Technology (PPGCTA/UFPB) and FAPESQ Notice no. 4/2021 PDPG from the Semi-Arid Region, Brazilian Agricultural Research Corporation (EMBRAPA, Project PROBIOLACT 23.16.05.038.00.00), Conselho Nacional de Desenvolvimento Científico e Tecnológico (CNPq, Projects 307075/2020-6 and 308253/2020-5), Coordenação de Aperfeiçoamento de Pessoal de Nível Superior (CAPES/PROAP), Fundação de Apoio à Pesquisa do Estado da Paraíba (FAPESQ, Projects 028/2018 and 2778/2023), Fundação Parque Tecnológico da Paraíba (PaqTcPB, Project AC-NUPEA-UEPB), Pró-Reitoria de Pesquisa (PROPESQ/PRPG/UFPB, internal productivity call 03/2020-PVM 13515-2020). This study was also financed by Paraíba State University, grant #01/2024. The funders had no role in study design, data collection and analysis, decision to publish, or preparation of the manuscript.

==============================
Background

Fruits are sources of bioactive compounds such as phenolics that bring health benefits to consumers. The addition of fruit products and microorganisms with probiotic potential in fermented goat milk can facilitate the acquisition of these benefits through diet. In this sense, the objective of this study was to evaluate the effect of incorporating a mixture of ingredients from jaboticaba (Myrciaria cauliflora), jambolana (Syzygium cumini), and mandacaru (Cereus jamacaru) fruits on fermentation parameters (pH, titratable acidity, viability of the native culture Lactiplantibacillus plantarum CNPC003 and the starter culture), associated with pigmentation (phenolic compound content and color) through experimental mixture design.

Methods

A simplex-centroid experimental design was conducted, comprised of seven trials totaling the addition of 30% of the fruit preparations in the final formulation of fermented milk and one control trial (without addition of preparations), with the response being the total phenolic content and the instrumental color parameter a*. Fermentations were carried out with the addition of the native culture Lactiplantibacillus plantarum CNPC003 and the starter culture Streptococcus thermophilus. Subsequently, analyses of pH, titratable acidity, viability of the native and starter cultures, total phenolic compound content, and the instrumental color parameter a* were performed.

Results

The final pH among trials ranged from 4.55 to 4.69, titratable acidity ranged from 0.59 to 0.64, the population of L. plantarum CNPC003 reached levels exceeding 8 log CFU/g, as did the population of Streptococcus thermophilus. The content of phenolic compounds was higher in trials T1, T5, and T7, as well as the color parameter (a*). The use of experimental mixture design contributed to the development of products with high viability of L. plantarum, high content of phenolic compounds, and a characteristic color of the added fruits, bringing benefits to consumer health.

Introduction

The increase in degenerative chronic diseases such as cancer, diabetes, obesity, chronic heart diseases, among others, is leading consumers to seek foods rich in fibers, vitamins, natural pigments, and phenolic content, as several studies demonstrate that these compounds can help prevent these diseases. Among these foods, there are fruits (Hernández-Carranza et al., 2019; Global Functional Foods Market, 2022; Granato et al., 2020), sources of bioactive compounds, such as total phenolics, which are important antioxidants with proven action in the trial and/or prevention of various diseases (Araújo et al., 2021). Moreover, by combining different fruits, it is possible to create products with a broad range of vitamins, minerals, antioxidants, and fibers, benefiting both their nutritional profile and the flavor. For example, smoothies that blend baobab and pineapple and black-plum increase the intake of vitamin C, calcium and magnesium, while yogurts and fermented milks with strawberry and acerola jam provide additional bioactive compounds. Furthermore, fruit blends in jams, such as strawberry, raspberry, and blueberry, boost the antioxidant content, such as anthocyanins, contributing to overall health (Adedokun et al., 2022; Morais et al., 2024).

However, a wide variety of tropical fruits, with potential benefits for nutrition and health, are underutilized or generate by-products such as peels, seeds, and pulps that are not used by the industry. Myrciaria cauliflora, known as jaboticaba, is a Brazilian fruit with high anthocyanins and phenolic content in its peel, which makes up 30% of the fruit massa. Although the fruit is widely used in the food industry for making sweets, jams, extracts, liqueurs, among others, the peel and seed together account for 50% of the fruit mass and are typically discarded (Neri-Numa et al., 2018; Quatrin et al., 2020; Senes et al., 2021). Syzygium cumini (L.) Skeels, or jambolana, also from the Myrtaceae family, has a distinctive astringent taste due to its organic acids and tannins, and although it has high anthocyanin levels compared to some other tropical fruits, it is less popular locally (Nascimento-Silva, Bastos & Da Silva, 2022; Santos et al., 2020; Peixoto et al., 2016). Cereus jamacaru, or mandacaru, is a cactus fruit noted for its high sugar content, fiber, and phenolic content, and its pulp is used in fermented goat milk to enhance its antioxidant properties and probiotic viability (Almeida et al., 2011; Fidelis et al., 2015; Araújo et al., 2021; Santos Neto et al., 2019; Soares et al., 2021; Galdino et al., 2020). In this sense, including a blend from such fruits into a new product could provide several benefits, such as increased nutrient (vitamins and minerals) content, improved flavor, and also enhanced antioxidant properties due to the diversity of bioactive compounds in the fruits. This kind of fruit combination could also support the viability of probiotic strains and boost consumer acceptance, since promising results were verified for dairy products in which these fruits were used solely (Almeida Neta et al., 2018; Galdino et al., 2020).

Considering the use probiotics, Lactiplantibacillus plantarum CNPC003 is native culture with notable technological and probiotic potential, either for use in dairy or in fruit-based products. This microorganism obtained viability higher than 7.60 log CFU/g after 6 h of fermentation of goat milk, when used sole or in co-culture with the starter Streptococcus thermophilus QGE for the fermentative acidification process, and in the presence or absence of inulin and/or fructooligosaccharides (Galdino et al., 2020). During 21 days of storage, L. plantarum CNPC003, used as a potential probiotic culture, was able to maintain viability close to 7 log CFU/g in skimmed fresh cheese (Oliveira et al., 2023) or higher this value in fermented desserts enriched with whey and jaboticaba peel (Almeida Neta et al., 2018). In Petit-Suisse cheese added with acerola pulp, it reached viability above 8 log CFU/g for 28 days of storage, being also able to survive under in vitro gastrointestinal simulation (Barcelos et al., 2020). Similarly, this culture maintained viable populations, above 6 log CFU/mL, after the in vitro digestion assay of a fermented non-dairy beverage with açaí berry on the 30th day of storage (Ribeiro et al., 2020). L. plantarum CNPC003 also showed proteolytic capacity, acidification and milk coagulation properties, diacetyl production (Moraes et al., 2017), as well as high exopolysaccharide (EPS) (Bomfim et al., 2020) and β-glucosidase outputs (Galdino et al., 2023), gathering the best characteristics of dairy technological suitability. Particularly, this ability as a β-glucosidase producer could ease the metabolism of polyphenols by this L. plantarum strain (Galdino et al., 2023).

Goat milk and its products offer benefits such as easy digestibility and lower allergenic potential due to the lower amount of β-lactoglobulin and α-s1-casein compared to bovine milk (Kielczewska et al., 2021). Thus, fermented goat milk can be an excellent vehicle for incorporating fruits with bioactive potential and probiotic candidates, being easy to include it into the daily diet. In this context, the study of the effect of adding ingredients from commercially underutilized fruits, sole or combined, allied to the functional properties of a goat milk fermented with a native probiotic candidate with potential of metabolizing polyphenols, can expand the knowledge about the behavior of native lactic cultures in the presence of fruit products for use in fermented goat products. This can provide innovation in the agribusiness sector, with special focus on small dairy producers, promoting the development of foods with multifunctional characteristics, and also contributing to the social, economic, and technological development of the regions involved in all the production chain. In view of these opportunities, the aim of this study was to evaluate the incorporation of a mixture of tropical fruit ingredients (jaboticaba, jambolana, and mandacaru) on the viability of the native culture L. plantarum CNPC003, the content of phenolic compounds, and the color of fermented goat milk through experimental mixture design.

Material & methods

Schematic overview of the experimental program

The schematic overview of the experimental program of the present study is presented in Fig. 1. Within this program, a simplex centroid design for experiments with mixtures was carried out with the aim of optimizing the development of a fermented goat milk added with the potentially probiotic native culture L. plantarum CNPC003, in coculture with the starter culture S. thermophilus QGE for the fermentative process, and the fruit ingredients jaboticaba, jambolana and mandacaru. Fermentative parameters (pH, titratable acidity, and lactic acid bacteria), as well as total phenolic content and color were used as dependent variables to evaluate the influence of the tropical fruit mixtures on the developed product. With this purpose, good laboratory practices were adhered to, as prescribed by the Centre of Research and Extension on Food (NUPEA) of the State University of Paraíba, Campina Grande, Brazil.

Figure 1 Flowchart representative of the experimental procedure.

Obtaining raw material

The fresh goat milk was provided by Carnaúba Farm, located in the municipality of Taperoá, Paraíba. Jaboticaba fruits (Myrciaria cauliflora) were harvested in the rural area of the municipality of Alagoa Nova (Geographical coordinates: 04′15″S 35°45′28″W), state of Paraíba, Brazil, in January 2021. The mandacaru (Cereus jamacaru) fruits were obtained from plant populations found spontaneously and dispersed in the Caatinga, in the municipality of Queimadas (Geographical coordinates: 7°21′28’″S 35°53′52″W), state of Paraíba, in May 2021. Jambolana (Syzygium cumini) originated from plants present spontaneously in the municipality of Campina Grande (geographical coordinates: 7°13′50″S 35°52′52″W), Paraíba, Brazil, in March 2021.

After harvesting, the fruits were selected, discarding those showing cuts, abrasions, and insect damage. Then, they were washed and sanitized in a sodium hypochlorite solution (200 mg/L of free chlorine) for 15 min. Subsequently, the pulp and peel fractions (mandacaru and jaboticaba) and pulp and seed (jambolana) were manually separated using stainless steel knives. The jaboticaba pulps and mandacaru peels, as well as the jambolana seeds, were not used in the present study.

The jaboticaba peels were kept in a solution of lemon juice in the ratio of 1:2:0.15 (peels:distilled water:lemon juice) for 45 min to reduce astringency, then filtered, and the liquid discarded.

Next, the fruit parts used in the present study, jaboticaba peel, jambolana pulp and mandacaru pulp and seed were processed (separately) in a blender (Philco, Multipro All In One PR + Citrus, Manaus, Brazil) for 10 min at 1,680 rpm. Afterwards, these ingredients were packed in low-density plastic bags and then subsequently immersed in a water bath at a 90 °C for 5 min, immediately cooled with an ice bath, and subsequently frozen at −18 °C, protected from light.

Activation process of Lactiplantibacillus plantarum CNPC003

The native strain L. plantarum CNPC003, provided in lyophilized form by Embrapa Goats and Sheep, Sobral, Ceará, Brazil, requiring activation before testing. This strain activation process was carried out according to methodology described by Barcelos et al. (2020), with modifications aimed at multiplying the cultures received, leaving them with greater metabolic activity and removing the material used in lyophilization (vehicle and other substances). This strain was chosen based on its ability to produce β-glucosidase, according to a study conducted by Galdino et al. (2023). The native strain was cultured in 5 mL of MRS broth, (Laboratório Conda S.A., Spain, distributed by Kasvi, São José dos Pinhais, Brazil) at 35 ± 2 °C for 24 h (first activation). Subsequently, a new cultivation step was performed by transferring 100 µL of the first activation to glass tubes containing 5 mL of MRS broth, which were then incubated in a bacteriological incubator (Nova Ética, model 402–4D, Vargem Grande Paulista, Brazil) at 35 ± 2 °C for 24 h (second activation). Then, the material was centrifuged at 3,500 rpm/15 min in a centrifuge (PARSEC, model CT–0603, Itajaí, Santa Catarina). The sediment (pellet) obtained was washed with five mL of 0.85% saline solution and centrifuged under the same previous conditions. The washed pellet was used in the fermentations.

Experimental design of fruit mixtures

To evaluate the effect of mixtures between the preparations of jaboticaba, jambolana, and mandacaru fruits added to the formulation, a simplex-centroid experimental design was conducted, always totaling 30% of the final formulation of fermented goat milk. The design for the three ingredients: jaboticaba peel (jaboticaba preparation (x 1)), jambolana pulp (jambolana preparation (x 2)), and mandacaru pulp and seed (mandacaru preparation (x 3)), comprised seven trials (T1 to T7) and one control trial (T8-without addition of fruit ingredients), as shown in Table 1. The response variables were the content of phenolic compounds, instrumental color parameter a*, and population of lactic acid bacteria. The trials were randomized, and statistical tests were applied using Statistica software version 7.0 (StatSoft, Tulsa, OK, USA).

Table 1 Experimental design of mixtures with coded variables (x1, x2 and x3 ) corresponding to the proportions of jaboticaba peel, jambolana and mandacaru pulps in the mixtures.

Trials	Independent variables	Response variables	
	x 1	x 2	x 3	Jaboticaba preparation (g/100 g)	Jambolana preparation (g/100 g)	Mandacaru preparation (g/100 g)	
T1	1	0	0	30	0	0	
T2	0	1	0	0	30	0	
T3	0	0	1	0	0	30	
T4	1/2	1/2	0	15	15	0	
T5	1/2	0	1/2	15	0	15	
T6	0	1/2	1/2	0	15	15	
T7	1/3	1/3	1/3	10	10	10	
Control	0	0	0	0	0	0	

Fermentations

For each treatment, 100 mL of milk was placed in a glass Erlenmeyer heat-treated at 110 °C for 15 s in a vertical autoclave (model AV75, Phoenix Luferco, Araraquara, Brasil). After being cooled to 42 °C, lyophilized starter culture Streptococcus thermophilus (QGE, Biotech Brazil Fermentos e Coagulantes Ltda, Alto Piquiri, Paraná) was added at a concentration of 0.002 g/100 mL, along with the sediment pellet (with an average of 7.5 × 108 UFC per pellet) of the native culture L. plantarum CNPC003 obtained after the second activation. When present, the fruit pulp preparations were added to the goat milk at a maximum ratio of 30 g/100 mL. The fermented milks were kept at 37 °C in an incubator oven (model Q0315M25, Quimis, Diadema, São Paulo, Brazil) for 6 h. Samples were collected in triplicate before (at zero time) and after 6 h of fermentation for pH, titratable acidity, and Lactobacillaceae population determination, and for instrumental color and total phenolic content determination only after 6 h of fermentation.

Evaluation of pH and titratable acidity

The pH and titratable acidity analyses were conducted before (at zero time) and after fermentation (6 h). The pH of the samples was determined using a pH meter (Kasvi, model K39-1420), following the analytical procedures described in method 017/IV of the Instituto Adolfo Lutz (2008).

The titratable acidity of the samples was evaluated according to the procedures described in method 426/IV of the Instituto Adolfo Lutz (2008) and expressed in terms of grams of lactic acid per 100 mL.

Enumeration of viable L. plantarum and Streptococcus thermophilus

The populations of the native culture of L. plantarum with probiotic potential were evaluated in triplicate. They were determined by adding 1 mL of each dilution (up to 10−7) to MRS medium acidified with acetic acid (Reagen, Quimiobrás, Rio de Janeiro, Brazil) to pH 5.4, followed by incubation at 37 °C for 48 h (Buriti & Saad, 2014; Pereira et al., 2019).

For the starter culture of S. thermophilus, the populations of this microorganism were determined by adding 1 mL of each dilution (up to 10−7) to M17 medium (Sigma-Aldrich, St. Louis, MO, USA) supplemented with lactose (Vetec–Química, Rio de Janeiro, Brazil) at a concentration of 5 g/L, followed by incubation at 37 °C for 48 h (Richter & Vedamuthu, 2001; Pereira et al., 2019).

Determination of total phenolic content

Extraction of phenolic constituents

This analysis was conducted according to Almeida Neta et al. (2018) on the fermented milks added with fruit ingredients just after 6 h of fermentation, comparing the content of phenolic between treatments.

For the preparation of phenolic extracts, portions of 1.25 g per sample were used and added to 5 mL of a methanol-HCl solution, which were left to rest in the dark at 4 °C overnight. The refrigerated samples were then centrifuged at a speed of 13,500 rpm for 5 min at 4 °C. Total depletion of phenolic content was achieved with a methanol-HCl solution, and this procedure was repeated four times. After each centrifugation, the supernatant was stored in a volumetric flask of 10 or 25 mL, depending on the sample, and adjusted with acidified methanol at the end of the extraction. An aliquot of 1.5 mL was taken for a microtube and then centrifuged for 1 min at 4 °C, with the supernatant collected for use in the analyses.

Determination of total phenolic content with Folin-Ciocalteu reagent

The content of phenolic content was performed in triplicate and in a dark environment, using the extract obtained by centrifugation, following the methodology of Karaaslan et al. (2011), and Dos Santos et al. (2017).

In 15 mL conical centrifuge tubes, 60 µL of the sample, 2,340 µL of distilled water, and 150 µL of Folin-Ciocalteu reagent (Sigma-Aldrich, St. Louis, MO, USA)) were added sequentially. The samples were shaken until completely homogeneous and left to stand for 8 min. Afterward, 450 µL of a 30% aqueous solution of sodium carbonate was added. The tubes were shaken again and left to stand for 30 min. Then, the absorbance was measured in a spectrophotometer at a wavelength of 750 nm, using acidified methanol to zero the equipment. The control test used in this analysis was prepared in the same way as described above except this one used 60 µL of acidified methanol in place of the sample. The results were expressed in milligrams of gallic acid equivalent per 100 grams of sample (mg GAE/100 g).

Color determination

The color determination was performed on all trials at the end of fermentation using a colorimeter (MiniScan HunterLab XE Plus, Reston, VA, USA), and the results were expressed in L* (lightness), ranging from 0 (black) to 100 (white); a*(green-red variation), ranging from -a* (green) to +a* (red); and b*(blue-yellow variation), ranging from -b* (blue) to +b* (yellow).); chroma (C*), calculated as the Eq. (1); and hue angle (h∘), calculated as the Eq. (2). (1) C∗=a∗2+b∗2

(2) h∘=tan−1a∗b∗.

For the experimental design, the variable used was ± a*, which corresponds to the variation in red coloration.

Statistical analysis

The results were expressed as mean ± standard deviation. The data for pH, titratable acidity, counts of L. plantarum and S. thermophilus, before and after fermentation, were subjected to non-parametric Wilcoxon analysis of variance. For the data on total phenolic compound content and color (L, a*, b*, C*, h∘), non-parametric Mann–Whitney U analysis of variance was used to verify the difference between trials, considering a significance level of p < 0.05. The correlation between the color parameter and total phenolic compound content was evaluated using the non-parametric Spearman test.

To evaluate the effects of the independent variables on the responses of L. plantarum viability, total phenolic compound content, and the +a* color parameter (red), response surface methodology was used when statistically significant differences were observed. In this sense, multiple regression models were proposed for each response. The quality of the models was evaluated by ANOVA and the Adjusted Coefficient of Determination (R2). Statistical analyses were performed using Statistica software, version 7.0 (Statsoft Inc., Tulsa, OK, USA).

Results

Effects on pH and titratable acidity

In Table 2, it is possible to observe the mean values of pH and titratable acidity obtained at the beginning (zero time) and after 6 h of fermentation in all trials according to the compositions defined by the experimental design in Table 1.

Table 2 Mean values and standard deviations of pH and acidity obtained at the at zero time and after 6 h of fermentation of goat fermented milk trials.

Trials	pH	Acidity (g lactic acid/100 g)	
	At zero time	After 6 h	At zero time	After 6 h	
T1	5.04 ± 0.032Af	4.66 ± 0.051Bab	0.29 ± 0.006Ba	0.60 ± 0.001Aa	
T2	5.85 ± 0.055Ac	4.64 ± 0.034Bb	0.25 ± 0.010Bcd	0.60 ± 0.010Aab	
T3	6.34 ± 0.036Ab	4.58 ± 0.041Bbc	0.25 ± 0.008Bcd	0.64 ± 0.047Aab	
T4	5.38 ± 0.038Ae	4.67 ± 0.051Bab	0.27 ± 0.007Bb	0.60 ± 0.001Aa	
T5	5.83 ± 0.030Ac	4.69 ± 0.017Ba	0.29 ± 0.013Ba	0.60 ± 0.002Aab	
T6	6.36 ± 0.055Ab	4.55 ± 0.010Bc	0.24 ± 0.047Bd	0.59 ± 0.014Ab	
T7	5.73 ± 0.029Ad	4.65 ± 0.049Bab	0.26 ± 0.006Bbc	0.60 ± 0.015Aab	
Control	6.70 ± 0.015Aa	4.56 ± 0.017Bc	0.25 ± 0.017Bd	0.61 ± 0.024Aab	
Notes.

A,BSuperscript capital letters in the same row denote significant differences between different sampling periods considering p <  0.05.

a, b, c, d, e, fSuperscript letters in the same column denote significant differences between different trials considering p < 0.05.

There was a significant decrease between (at zero time) pH (6.70–5.04) and after 6 h of fermentation (4.55–4.69) (p = 0.000380). Comparing the different trials there were significant differences for the most trials (p = 0.049535) at zero time pH (achieving the maximum value of 6.70 ± 0.015 for control trial, and minimum value of 5.04 ± 0.032 for T1 trial with 30 g/100 g of jaboticaba preparation), except between T3 (30 g/100 g of mandacaru preparation, pH value of 6.34 ± 0.036) and T6 (15 g/100 g of each jambolana and mandacaru preparation, pH value of 6.36 ± 0.055) and between T2 (30 g/100 g of jambolana preparation, pH value of 5.85 ± 0.055) and T5 (15 g/100 g of each jaboticaba and mandacaru preparation, pH value of 5.83 ± 0.030). At the end of fermentation there were significant differences between the most trials and the control (p = 0.049535), except for T3 (pH value of 4.58 ± 0.041) and trial T6 (pH value of 4.55 ± 0.010), which did not differ significantly from the control (pH value of 4.56 ± 0.017). Trial T1 (pH value of 4.66 ± 0.051) differed significantly (p = 0.049535) from T6 and the control but did not differ significantly from the others (p > 0.05).

For titratable acidity, there was a significant between zero time and 6 h of fermentation process in all trials (p = 0.000000). When comparing the trials at the same time, T2 (30 g/100 g of jambolana, 0.25 ± 0.010 g lactic acid/100 g), T3 (30 g/100 g of mandacaru, 0.25 ± 0.008 g lactic acid/100 g), and T6 (15 g/100 g of each jambolana and mandacaru, 0.24 ± 0.047 g lactic acid/100 g), without jaboticaba, did not differ significantly from the control sample before the fermentation process (at zero time). At the end of fermentation (6 h), trials T1 (30 g/100 g of jaboticaba, 0,60 ± 0,001 g lactic acid/100 g) and T4 (15 g/100 g each for both jaboticaba and jambolana, 0.60 ± 0.001 g lactic acid/100 g), only differed significantly from T6 (p = 0.049535).

Effects on viability of L. plantarum and Streptococcus thermophilus

Table 3 shows the averages of L. plantarum and S. thermophilus populations before (at zero time) and after 6 h of fermentation process in goat milk added with fruit ingredients. There was a significant increase in the populations of both L. plantarum (starting from 7.55–7.84 CFU/mL and ending with 8.00–9.24 CFU/mL) and S. thermophilus (starting from 7.61–7.85 CFU/mL and ending with 8.51–8.84 CFU/mL) when comparing the final and the initial times for all trials (p = 0.026876).

Table 3 Mean values and standard deviation ml of the population of L. plantarum and S. thermophilus obtained at the at zero time and after 6 h of fermentation of goat fermented milk trials.

Trials	L. plantarum (log UFC/mL)	S. thermophilus (log UFC/mL)	
	At zero time	After 6 h	At zero time	After 6 h	
T1	7.84 ± 0.04Ba	9.00 ± 0.59Aac	7.84 ± 0.04Ba	8.74 ± 0.25Aab	
T2	7.83 ± 0.016Ba	8.98 ± 0.63Aab	7.85 ± 0.10Ba	8.71 ± 0.10Aab	
T3	7.68 ± 0.10Bbd	9.24 ± 0.87Aab	7.80 ± 0.01Ba	8.69 ± 0.10Aab	
T4	7.59 ± 0.02Bcde	8.94 ± 0.23Aa	7.72 ± 0.15Bab	8.84 ± 0.21Aab	
T5	7.76 ± 0.02Bb	8.74 ± 0.90Aab	7.66 ± 0.20Bab	8.70 ± 0.17Aab	
T6	7.76 ± 0.16Babe	8.86 ± 0.81Aab	7.82 ± 0.12Ba	8.82 ± 0.10Aa	
T7	7.60 ± 0.03Bce	8.34 ± 0.27Abc	7.81 ± 0.05Ba	8.51 ± 0.30Aab	
Control	7.55 ± 0.03Bc	8.00 ± 0.35Ab	7.61 ± 0.01Bb	8.68 ± 0.03Ab	
Notes.

A,BSuperscript capital letters in the same row denote significant differences between different sampling periods considering p < 0.05.

a, b, c, d, e, fSuperscript letters in the same column denote significant differences between different trials considering p < 0.05.

When comparing the control at the final fermentation time (6 h), that achieved viabilities of 8.00 ± 0.35 CFU/mL and 8.68 ± 0.03 CFU/mL for L. plantarum and S. thermophilus, respectively, with the other trials, only trials T1 (9.00 ± 0.59 log CFU/mL) and T4 (8.94 ± 0.23 log CFU/mL) differed significantly (p = 0.049535) for the population of L. plantarum CNPC003, and only trial T6 (8.82 ±0.10 log CFU/mL) differed for the population of S. thermophilus.

Effects on total phenolic compounds

The contents of phenolic compounds were 434.67 ± 32.88 mg GAE/100 g, 158.58 ± 12.48 mg GAE/100 g, and 115.40 ± 3.10 mg GAE/100 g for the jaboticaba, jambolana, and mandacaru preparations, respectively. After the fermentation of goat milk with these preparations and their combinations, there were significant differences (p = 0.049535) among all fermented milk trials for the total phenolic content after 6 h of the fermentation process (Table 4). Comparing all trials with the control, a significant increase (p = 0.049535) in phenolic content was observed due to the addition of fruit preparations to the fermented milk. After addition to the milk and fermentation process, it was observed that trial T1, containing 30 g/100 g of jaboticaba peel, obtained the highest significant value (p = 0.049535) of phenolic content (224.83 ± 2.36 mg GAE/100 g). Even at concentrations below 30 g/100 g, the trials that had jaboticaba peel in their composition, T4, T5, and T7, showed the highest levels of phenolic compounds, with 146.20 ± 8.08 mg GAE/100 g, 90.12 ± 9.37 mg GAE/100 g, and 76 ± 1.54 mg GAE/100 g, respectively.

Table 4 Mean values and standard deviations of phenolic content and color parameters of goat fermented milk added to fruit ingredients after 6 h of fermentation.

Trials	Total phenolic content (mg GAE/100 g)–after 6 h of fermentation	Color (after 6 h of fermentation)	
		L	a*	b*	C*	h o	
T1	224.83 ± 2.36a	20.76 ± 0.07g	7.38 ± 0.07a	−2.71 ± 0.02f	7.86 ± 0.07a	339.84 ± 0.10c	
T2	46.67 ± 0.82e	25.65 ± 0.16e	4.81 ± 0.04d	−3.67 ± 0.05g	6.05 ± 0.04c	322.63 ± 0.50d	
T3	37.45 ± 2.10f	26.21 ± 0.07d	0.14 ± 0.05f	2.62 ± 0.10b	2.62 ± 0.06h	86.94 ± 1.24f	
T4	146.20 ± 8.08b	20.12 ± 0.11h	5.36 ± 0.10c	−1.92 ± 0.08e	5.70 ± 0.02d	340.31 ± 0.60c	
T5	90.12 ± 9.37c	21.21 ± 0.18f	4.82 ± 0.30d	−1.34 ± 0.65d	5.00 ± 0.30e	344.50 ± 0.27b	
T6	22.38 ± 0.92g	28.27 ± 0.09c	3.85 ± 0.05e	0.58 ± 0.05c	3.90 ± 0.04f	8.57 ± 0.89g	
T7	77.64 ± 1.78d	31.54 ± 0.06b	6.17 ± 0.06b	−1.20 ± 0.06d	6.29 ± 0.05b	348.94 ± 0.50a	
Control	7.25 ± 0.48h	35.76 ± 0.07a	−0.25 ± 0.02g	2.73 ± 0.02a	2.73 ± 0.023g	95.23 ± 0.34e	
Notes.

a, b, c, d, e, f, g, hSuperscript letters in the same column denote significant differences between different trials considering p < 0.05.

Effects on products’ color

The obtained values from the instrumental color determination obtained for the fermented milk trials are presented in Table 4.

Lightness (L) differed significantly in all trials (p = 0.049535). The L* values, ranging from 20.12 for the T4 trial with 15 g/100 g of each jaboticaba and jambolana preparations to 35.75 for the control trial, decreased significantly (p = 0.049535) as the concentrations of ingredients with jaboticaba and/or jambolana increased.

For the paramenter -a* +a* (-a* green; +a* red), trials T2 and T5 did not differ significantly from each other (p = 0.512691); however, they differed from the other trials (p = 0.049535) Regarding the values of a* (red), there was a significant increase (p = 0.049535) among the trials when adding jaboticaba peel and/or jambolana pulp compared to trials with the addition of mandacaru pulp. Trials T1(7.38 ± 0.07), T7 (6.17 ± 0.06), and T4 (5.36 ± 0.10) showed a more reddish coloration and the highest values of this parameter, T2 (4.81 ± 0.04), T5 (4.82 ± 0.30), and T6 (3.82 ± 0.05) trials had a more pinkish coloration, while trial T3 (0.14 ± 0.05) was white. For the parameter -b*+b* (-b* blue; +b*yellow), trials T5 (−1.34 ± 0.65) and T7 (−1.20 ± 0.06) did not differ significantly from each other (p = 0.126631). The values of -b*+b* (ranging from -3,67 for T2 trial to 2.73 for control trial), in contrast to the values of a*, decreased significantly (p = 0.049535) in trials with the presence of jaboticaba peel and/or jambolana pulp. For the color parameter C* (chroma saturation), all trials differed significantly from each other (p = 0.049535), with the lowest (2.62 ± 0.06) and the highest (7.86 ± 0.07) values obtained for T3 and T1 trials, respectively. For the hue angle (h∘), only trials T1 and T4 did not differed significantly from each other (p = 0.512691). Trials T1 and T4. Trials T1(339.84 ± 0.10), T2(322.63 ± 0.50), T4(340.31 ± 0.60), T5(344.50 ± 0.27), and T7 (348.94 ± 0.50) obtained higher h∘ values due to the presence of jaboticaba and/or jambolana in the composition, presenting a purple-reddish coloration, opposed to the T3 (86.94 ± 1.24), T6 (8.57 ± 0.89) and control (95.23 ± 0.34) trials. In the present study, the Spearman correlation between color (a*) and the content of phenolic compounds was significantly (p < 0.05) positive, with an rs value of 0.834, demonstrating that both variables increased simultaneously.

Table 5 Coefficients (x) with their respective standard errors of the quadratic model analysis adjusted to the experimental data of phenolic compounds content of the treatments of fermented goat’s milk after 6 h of fermentation.

	Values	
Prepared jaboticaba (x1)	225.04 [2.79]	
Prepared jambolana (x2)	46.88 [2.78]	
Prepared mandacaru (x3)	37.66 [2.78]	
Prepared jaboticaba plus Prepared jambolana (x12)	37.66 [12.81]	
Prepared jaboticaba plus mandacaru (x13)	−168.16 [12.81]	
Prepared jambolana plus mandacaru (x23)	−82.82 [12.81]	
Prepared jaboticaba plus jambolão plus mandacaru (x123)	<0.05	
F (model)	783.23	
p (model)	0.000000	
Degree of freedom (model)	5	
Degree of freedom (total)	20	
p (lack of fit)	0.42	
R 2	0.9962	
Ajusted R 2	0.9949	

Optimization via regression models

Quantitative relationships between variables and responses can be defined by regression models, covering the entire tested experimental range and possible interactions. The linear regression model was tested for all target responses (dependent variables) at the significance level of 5%, using the results after 6 h of fermentation. When the linear polynomial models showed low coefficient of determination (R2) and significant lack of fit (p < 0.05), more complex models such as quadratic or cubic regressions were attempted. The choice of the most appropriate model was based on lack of fit and coefficient of determination. For the parameter viability of L. plantarum, none of the models were significant, and therefore, this parameter was not used in the optimization (Data S1). In Table 5, it is observed the coefficients and their respective standard errors from the analysis of the quadratic model adjusted to the total phenolic content. This model was significant (p = 0.00000), obtained a non-significant lack of fit (p = 0.42), suggesting a good fitness of the model to the experimental data, with R2 0.99. In this sense, the predictive model explains 99% of the variation obtained for the phenolic content response. Based on this model, it was possible to obtain the Eq. (3) for the total phenolic content. (3) y ˆ1=225.04x1+46.88x2+37.65x3+37.66x1x2−168.15x1x3−82.82x2x3

where y ˆ(1) is the total phenolic content.

Figure 2 Triangular diagram of total phenolic content (mg GAE/100 g) in fermented goat’s milk after 6 h of fermentation (x1= jaboticaba, x2= jambolana, x3= mandacaru).

Based on the predictive model for the total phenolic content response, there was a significant interaction when comparing the mixture of two fruit ingredients, but there was no positive interaction in the mixture of all three fruit preparations. Based on Fig. 2, it is observed that as the concentration of jaboticaba peel increases, there is an increase in the total phenolic content. The analysis of variance of the cubic model adjusted to the color parameter a* of the fermented goat milk added with fruit preparation is observed in Table 6. As obtained for the total phenolic content, the model for the color parameter a* was significant (p = 0.00000), obtained a lack of fit (p = 1.0000) with R2 0.99 and, in this sense, the predictive model explains 99% of the variation in the response of color parameter a*. Therefore, the special cubic regression model for the color parameter a* in goat milk added of fruit ingredients after 6 h of fermentation is represented by the Eq. (4). (4) y ˆ2=7.38x1+4.81x2+0.14x3−2.92x1x2+4.92x1x3+5.50x2x3+33.16x1x2x3

where y ˆ (2) is the parameter for color a*.

Regarding the color parameter a*, all coefficients were significant, including the interaction of the three fruits. Based on the Fig. 3, that presents the contour plot of this model, the development of a product with a more reddish color is possible with the addition of the jaboticaba, jambolana, and mandacaru preparations.

Discussion

Effects on fermentative and pigmentation parameters

At the end of the fermentation process, there was a decrease in pH and an increase in titratable acidity (Table 2). This is due to the ability of lactic acid bacteria S. thermophilus and L. plantarum to produce acids during this process, which results in significant effects on processing time and economic viability of milks (Tian et al., 2017; Liang et al., 2024). In this sense, the simultaneous addition of a starter culture, such as S. thermophilus, and an adjunct culture, such as L. plantarum, are more effective for the acid production, particularly lactic acid, aiming to achieve the proper acidity in a shorter time, compared to the use of each culture separately (Galdino et al., 2021). In parallel, the formation of these acids favors the precipitation of proteins, forming the curds that give a viscous structure to fermented milks and the formation of the characteristic aroma of these products (Gu et al., 2021).

Table 6 Coefficients (x) with their respective standard errors of the quadratic model analysis adjusted to the experimental data of color a* in the fermented goat milk after 6 h of fermentation.

	Values	
Prepared jaboticaba (x1)	7.38 [0.038]	
Prepared jambolana (x2)	4.81 [0.038]	
Prepared mandacaru (x3)	0.14 [0.038]	
Prepared jaboticaba plus jambolana (x12)	−2.92 [0.187]	
Prepared jaboticaba plus mandacaru (x13)	4.92 [0.187]	
Prepared jambolana plus mandacaru (x23)	5.5 [0.187]	
Prepared jabuticaba plus jambolão plus mandacaru (x123)	33.16 [1.32]	
F (model)	3578.75	
p (model)	0.000000	
Degree of freedom (model)	6	
Degree of freedom (total)	20	
p (lack of fit)	1.0000	
R 2	0.9993	
Adjusted R2	0.9990	

Figure 3 Triangular diagram of color a∗ in fermented goat’s milk after 6 h of fermentation (x1= jabuticaba, x2= jambolana, x3= mandacaru).

Similarly, the presence of organic acids in fruits can directly influence the characteristics of products made with fruit ingredients, such as pH and titratable acidity. According to the literature, the pH values of jaboticaba, jambolana, and mandacaru are 3.25, 4.12, and 3.93, respectively (Soares et al., 2021; Santos et al., 2020; Rosa et al., 2023). This fact explains the significant differences (p = 0.049535) before fermentation in all trials of this study, with the trials containing the jaboticaba ingredient, even at different concentrations, tending for the lower pH values and higher titratable acidity (Table 2). Additionally, the different concentrations of each fruit ingredient used (Table 1) contributed, in part, to the significant difference (p = 0.049535) in final pH and titratable acidity after 6 h of the fermentation process, considering that, at this point, the influence from the metabolism of the lactic cultures is already present, rendering also significant lower pH values for formulations with mandacaru, either sole or combined with jaboticaba. The positive stimuli or not from fruit ingredients present in dairy matrices, including fermented milks, on the metabolism of starter cultures and probiotic adjuncts is dependent on the composition of carbohydrates, organic acids, phenolic compounds, vitamins, among others (Barcelos et al., 2020; Borgonovi et al., 2022). Particularly in fermented dairy products, the benefits of adding fruit ingredients, beyond to speed up the acidification process, shortening the fermentation time, could include the improvement in the viability of starter and probiotic cultures in the final product during their refrigerated storage (Almeida Neta et al., 2018; Borgonovi et al., 2022).

Nonetheless, despite, some carbohydrates and organic acids can be used as energy sources by lactic acid bacteria, improving their population growth, while antioxidant vitamins and phenolic compounds may help them to prevent oxidative metabolism, extending their survival, the incorporation of fruit juices or pulps to probiotic dairy products, sometimes may affect negatively the survival of probiotic bacteria (Barcelos et al., 2020). In the present study, the significant increase (p < 0.05) in the population of lactic acid bacteria after 6 h of fermentation in all trials of the present study (Table 3) demonstrated that the added fruit ingredients did not result in a negative effect on the population of these bacteria. Similarly, Borgonovi, Casarotti & Penna (2021) observed that the pulps of buriti and passion fruit did not negatively interfere with the viability of lactic acid bacteria.

Within this context, it is important to emphasize that one of the great challenges of the food industry in using probiotics in various food matrices is their low survival, depending on extrinsic and intrinsic parameters such as pH, temperature, processing, and interaction with other ingredients added to the formulation (Ghasempour et al., 2019). To exert beneficial physiological effects, probiotic strains must be viable at a minimum concentration of 6 log CFU/mL in the final product and throughout its shelf life for daily consumption of 100 g of product, although this beneficial quantity may vary depending on the strain and the product elaborated (Costa et al., 2019; Guimarães et al., 2020). In this sense, the food matrix should favor and ensure this quantity until the product is consumed. In the present study, the population of lactobacilli at the end of fermentation remained above 8 log CFU/mL (Table 3). Moreover, since the significant lowest L. plantarum viability was verified for the control trial (Table 3), without the addition of fruits, the fruit preparations of the present study actually improved the growth of this culture in the fermented milks. As the T1 and T4 trials, both of them with the jaboticaba preparation, were the products that had significant high L. plantarum viability compared to the control, part of this improvement could be related to the capability of metabolize the phenolic compounds. In a previous study of the present research group, L. plantarum CNPC003 was able to produce β-glucosidase, characteristic that is closely linked to the ability of metabolize phenolic compounds (Galdino et al., 2023).

Considering the total phenolic content observed in the fruit ingredients of the present study, the jaboticaba preparation, in fact, exhibited the highest value (434.67 mg GAE/100 g), nearly three times higher than that observed for the jambolana preparation (158.58 mg GAE/100 g,) and almost four times higher than that observed for the mandacaru preparation (115.40 mg GAE/100 g). In parallel, the total phenolic content observed for the jambolana preparation was nearly one-third higher than that observed for the mandacaru preparation. Within this respect, Wu, Long & Kennely (2013) reported approximately 460 mg GAE/100 g of phenolic compounds in jaboticaba, much higher compared to the phenolic content of mandacaru and jambolana fruits. However, Sousa et al. (2021) reported a phenolic content for jaboticaba peel syrup of 148.12 mg GAE/100 g, and 2,388.54 mg GAE/100 g for the hydroalcoholic extract of jaboticaba peel. On the other hand, Faria, Marques & Mercadante (2011) and Carvalho et al. (2017) reported the phenolic content of jambolana as 148.3 mg GAE/100 g and 122 mg GAE/100 g, respectively. The phenolic content of mandacaru fruit was 115.40 mg GAE/100 g, a value close to that reported by Soares et al. (2021) of 107.31 mg GAE/100 g.

Like the results of the present study (Table 4) with fermented milk assays, Almeida Neta et al. (2018) reported an increase in phenolic content when incorporating jaboticaba pulp into fermented dessert with goat whey. Similarly, Garcia et al. (2020) and Dantas et al. (2022) also reported higher total phenolic content after adding jambolana pulp to a lactose-free fermented dairy beverage.

In the present study, the difference in phenolic content among the trials is due to the different concentrations of fruit preparations (Table 4). It is observed that trial T1, containing 30% jaboticaba peel, obtained the highest significant value (p < 0.05) of phenolic content (224.83 ± 2.36 mg GAE/100 g). Even at concentrations lower than 30%, trials containing jaboticaba peel showed the highest phenolic compound content, T4, T5, and T7, with 146.20 ± 8.08 mg GAE/100 g, 90.12 ± 9.37 mg GAE/100 g, and 76 ± 1.54 mg GAE/100 g, respectively. Thus, jaboticaba peel stood out and can be well utilized for addition to fermented milk, aiming to provide high phenolic values and ensuring optimal fruit utilization.

Color is a crucial attribute in the food industry, influencing consumer acceptability. Undesirable coloration can lead to poor product acceptance and reduced market value (Freitas-Sá et al., 2018). The results obtained for luminosity (L*) in this study (Table 4), where 100 represents white and zero represents black (Costa et al., 2015; Chudy et al., 2020), suggest that the pulps decreased the luminosity of the fermented milk, which may be related to the red-purple coloration of jaboticaba peel and jambolana pulp. According to Spence (2023), consumers associate red color with sweetness, suggesting that trials with more intense red coloration would be better accepted. In this regard, the increase in the +a* color parameter (Table 4), indicates that the presence of compounds such as anthocyanins found in jambolana (Nascimento-Silva, Bastos & Da Silva, 2022) and jaboticaba (Quatrin et al., 2020; Neri-Numa et al., 2018) favors the reddish coloration of the final product.

Hernández-Carranza et al. (2019) demonstrated that the addition of forage palm fruit peel, which has a reddish color, decreased the luminosity, and increased the red color +a* of the formulated yogurts, while the b* color parameter was not affected by the addition of this by-product.

Chroma saturation (C*) is related to the intensity or amount of a hue and indicates the proportion in which it is mixed with white, black, or gray, allowing to distinguish between strong and weak colors (Chudy et al., 2020). Therefore, the trials that showed a stronger hue were those that had jaboticaba and/or jambolana in their composition (Refer to Table 4).

The h∘ characterizes the color quality (red, green, blue, and yellow), making it possible to differentiate between colors (Chudy et al., 2020). In this study, the highest h∘ values were obtained for products containing jaboticaba and/or jambolana preparations (Refer to Table 4).

Interactive potentials of parameters

The Spearman correlation between color (a*) and phenolic content was significantly (p < 0.05) positive with a value of rs of 0.834, demonstrating that both variables increased simultaneously. A correlation with rs values above 0.8 is considered a strong correlation (Akoglu, 2018). It can be inferred, therefore, that as the color a* increased, so did the phenolic content in the fermented goat milk trials developed in this study.

Based on the predictive model for the phenolic content response, there was a significant interaction when comparing the mixture of two fruit ingredients, but there was no positive interaction in the mixture of all three fruit preparations (Refer to Table 5). Analyzing Fig. 2, it is possible to infer that jaboticaba peel is the ingredient that has the most influence on the phenolic content, confirming the previously discussed color results. Although the interaction of the three fruit ingredients was not significant regarding the phenolic compound content, such interaction could provide a greater diversity of bioactive compounds in this product.

Regarding the parameter of color a*, all coefficients were significant, including the interaction of the three fruits (refer to Table 6 and Fig. 3). Obtaining a product with a more reddish color is possible with the addition of jaboticaba, jambolana, and mandacaru preparations.

In this sense, choosing the three fruits for the preparation of a fermented product allows for combining the phenolic composition present in all three fruits and at the same time promotes an appealing color for consumers.

Conclusions

The population of lactic acid bacteria, including the starter culture S. thermophilus and the native L. plantarum, remained above 8 log CFU/mL in all trials, demonstrating that the pulps neither negatively interfere in the growth of populations of these microorganisms nor with the acidification capacity of them in coculture during the fermentation process. Particularly for L. plantarum, achieving populations generally recommended for probiotic products, a stimulatory potential of the fruits was observed, mainly for jaboticaba, which deserves to be investigated with more details in futures studies together with the evaluation of phenolic compounds profile, antioxidant capacity and sensory acceptability.

As a result of the highest content of total phenolic content in the jaboticaba peel preparation compared to the other fruits, the fermented milks with the presence of the jaboticaba ingredient showed a higher content of phenolics. There was a positive correlation between the total phenolic content and the color parameter +a* (red) of the fermented milks studied. As consequence, the presence of jaboticaba in the fermented milks resulted in a higher total phenolic content, as well as a more reddish coloration. The mandacaru ingredient, with both lower phenolic content and *a values compared to the other fruits, had a smaller contribution for these parameters in the fermented milks.

The present study, using the simplex centroid experimental design, contributed to providing different combinations of these fruits, making possible to understand their relationship with the high levels of phenolic compounds and a redder color, which are associated with attractive products. These advantages in fermented milk manufacture, allied to the presence of a potential probiotic culture, may promote the consumption of products that benefit the consumer health.

Supplemental Information

Supplemental Information 1 Raw data

The experimental planning carried out taking into account the phenolic content and color, in addition to the viability data of Lactobacillaceae, pH and acidity, Streptococus thermopilus and the color parameters: L, a, b, HUE and phenolic content of the fruit preparations.

The authors thank the Carnauba farm for the donation of goat’s milk and Biotech Fermentos e Coagulantes for the donation of the starter culture.

Additional Information and Declarations

Competing Interests

Author Contributions

Patent Disclosures

Data Availability

Karina Maria Olbrich do Santos and Antonio Silvio do Egito are employees of Brasilian Agricultural Research Corporation.

Isadora Kaline Camelo Pires de Oliveira Galdino analyzed the data, prepared figures and/or tables, authored or reviewed drafts of the article, and approved the final draft.

Gabriel Monteiro da Silva performed the experiments, authored or reviewed drafts of the article, and approved the final draft.

Miqueas Oliveira Morais da Silva performed the experiments, authored or reviewed drafts of the article, and approved the final draft.

Giordanni Cabral Dantas analyzed the data, authored or reviewed drafts of the article, and approved the final draft.

Elainy Virgínia dos Santos Pereira analyzed the data, authored or reviewed drafts of the article, and approved the final draft.

Tiago Almeida de Oliveira analyzed the data, authored or reviewed drafts of the article, and approved the final draft.

Karina Maria Olbrich dos Santos conceived and designed the experiments, authored or reviewed drafts of the article, and approved the final draft.

Antonio Silvio do Egito conceived and designed the experiments, authored or reviewed drafts of the article, and approved the final draft.

Flávia Carolina Alonso Buriti conceived and designed the experiments, analyzed the data, prepared figures and/or tables, authored or reviewed drafts of the article, and approved the final draft.

Haíssa Roberta Cardarelli conceived and designed the experiments, analyzed the data, authored or reviewed drafts of the article, and approved the final draft.

The following patent dependencies were disclosed by the authors:

Galdino, I. K. C. O.; Buriti, F. C.; Dos Santos, K. M. O.; Do Egito, A. S. 2020. BR 10 2020 023147 2. Depósito: 12 nov.

The following information was supplied regarding data availability:

The raw measurements are available in the Supplementary File.

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
