# Peer review of "Effects of tropical fruit blends on fermentative and pigmentation aspects of probiotic native cultured goat milk"

_PeerJ, doi:10.7717/peerj.18813_

## Round 0.1 · original submission · Major Revisions

Thank you authors for your patience. You can see reviewers have raised ample concerns that need your kind attention. Also, I have some observations below, which I encourage you to attend to in great detail:
a) The title of this work is too long, it could be shortened to: "Effects of tropical fruit blends on fermentative and pigmentation aspects of probiotic native cultured goat milk"
This is because the word "fermentative implicitly depicts there is fermentation occurring in the food technological process ok...after reading the work, there is a lot of statistical effects, so it is best to use effects in the title ok

b) In the introduction, firstly, please expand paragraph 1 having introcuced importance of fruits, talk about in same paragraph the making of fruit blends, and how it further improves the overall nutritional value of emergent product.
Then, please merge paragraphs 2,3 and 4 as one, and further condense it strictly to important points. (remove things like, 'the study conducted by', 'observed that...,' 'some studies demonstrate....go straight to the point
Please, merge paragraphs 5, 6, 7 and 8 as one and further condense them. Apply your discretion to clearly justify this research, why? how?

c) Materials and methods, please create a new subsection to start this, called 'Schematic overview of the experimental program', which should comprise max 5 sentences, and supported by a flow diagram that depicts the key experimental stages. Sentence 1 should have title of the flow diagram figure, sent 2 connects it with the objective of this work, sent 3, 4 and 5 will reiterate most important sections, ending with good laboratory practices were adhered to as prescribed by Department of....., University of....

d) Please, in the results section, you are no more determining these parameters, you are showing the effects on them...please subtitles should be "Effects on pH and titratable acidity (not jiust acidity), Effects on fermenting microbes, Effects on total phenolic compounds, Effects on color, Optimization via regression models (Please for optimization, delete obtaining regression models. Please ensure that these subsections are all one paragraph only. For example, optimization has >6 paragraphs, let it be max 2 ok

e) Discussion needs some work. You have 6 tables and twp figures (which would in revised manuscript become figures 2 and 3),,,so, in your discussion, please at all the places where specific data from each table or figure is presented, please indicated (Refer to Table x) or (Refer to Figure x). Make sure that all the Tables and Figures mentioned in the results are captured in the discussion
Remember in the results,discussion and conclusion , it is not acidity, it is titratable acidity, be consistent with your data ok

f) In the discussion, please address the following contexts:
- Give more emphasis on the "how", and "why," and not just "what" the literature has said, and what your results have shown, please go more indepth in the discussion, and less on describing the data.

For a statistically significant outcomes, please show in the bracket the specific p-value, and the F-change or R-sq(adj) values). You used statistica, so it must have provided you this data. It is important for you to show this, ok

Same applies to your regression model, and correlation, provide the detail of the statistics, you cannot just mention low coefficient of determination (r), and significant lack of fit (p<0.05)...no, please provide the speciific value, you have them, show them in the text, even if they are shown in the table/figure, reiterate them in the text.

Discussion should be divided into two major parts:

Effects on fermentative and pigmentation parameters

Interactive potentials of parameters

Kindly make effort to have another look at your results, and pull out the various places where the effects/trends potentially connect ...remember you did correlation, and optimization, so you must have various aspects of the effects on parameters behaving similarly, or oppositely.
The word 'potentials' is useful to indicate that the behaviours.trends are suggested, yet observed, which lays foundation for future studies to confirm

g) Your conclusion is quite scant, please reflect on your findings, which trends/effects on parameters are most striking and why, and which is least, and why?
Which aspects of your findings creates some doubt, in your opinion, and why? Would or should it be the basis for future studies? You can make some recommendations here as well for future studies . I have a strong feeling that some aspects of your study are novel perspectives, so, please place some emphasis on this here ok

Look forward to your revised manuscript. Thank you for your kind efforts.

Reviewer 1 ·

Basic reporting

In the article titled "Influence of tropical fruit blends on fermentative parameters and pigmentation in fermented goat milk with potentially probiotic native culture", it is understood that the effects of 3 different regionally valuable fruits on some quality parameters in goat milk fermented with probiotic microorganisms in combination were studied. For this purpose, Total Phenolic Substance, pH and titratable acidity, color values ​​and the viability of L. plantarum and Streptococcus thermophilus were determined in the product. At the same time, fruit mixtures were determined using the experimental design method.
Although the aim of the study was good, the characteristics of the product could not be fully reflected due to the few quality analyzes performed. Examining the study in terms of total monomeric anthocyanin and in vitro antioxidant activity will add more meaning and enrich the study.

Keywords:
The number of keywords is not sufficient to represent the entire text. Therefore, keywords should be kept broad enough to reflect the entire study (5-6 words may be sufficient).

In Introduction:
“Among these foods, we have fruits (Hern·ndez-Carranza et al., ….” The subject “we” is used in this expression. Care should be taken not to use general expressions with personal subjects.

What is the purpose of combining three different fruits used in the study? It would be useful to mention this in the purpose section.

Experimental design

In Materials and Methods:
The structure of the fruits used in the study also contains anthocyanins, which are flavonoids with antioxidant properties. Total Monomeric Anthocyanin and Antioxidant capacity analyzes will also contribute to the study in terms of how stable the fermented goat milk remains in its structure and how it contributes to antioxidant activity.

Validity of the findings

The conclusion section should be further developed regarding the findings obtained and the advantages of the experimental design method applied.

All underlying data have been provided; they are robust, statistically sound, & controlled.

Reviewer 2 ·

Basic reporting

See attached file

Experimental design

See attached file

Validity of the findings

See attached file

Additional comments

See attached file

Annotated reviews are not available for download in order to protect the identity of reviewers who chose to remain anonymous.

---

## Round 0.2 · accepted · Accept

Thank you authors for revising your work, and improving the quality. It is acceptable for publication. Thank you for your scholarly contribution to PeerJ, and look forward to your future scholarly contributions.